# Role of *SiPHR1* in the Response to Low Phosphate in Foxtail Millet via Comparative Transcriptomic and Co-Expression Network Analyses

**DOI:** 10.3390/ijms241612786

**Published:** 2023-08-14

**Authors:** Guofang Xing, Minshan Jin, Peiyao Yue, Chao Ren, Jiongyu Hao, Yue Zhao, Xiongwei Zhao, Zhaoxia Sun, Siyu Hou

**Affiliations:** 1College of Agriculture, Shanxi Agricultural University, Jinzhong 030801, China; sxauxgf@126.com (G.X.); jms1995mashy@163.com (M.J.); zhaoxiasun@sxau.edu.cn (Z.S.); 2Hou Ji Laboratory in Shanxi Province, Shanxi Agricultural University, Taiyuan 030031, China; 3College of Life Sciences, Shanxi Agricultural University, Taigu 030801, China

**Keywords:** co-expression network, comparison transcriptomic, functional analyses, foxtail millet, low Pi stress, PHR1

## Abstract

Enhancing the absorption and utilization of phosphorus by crops is an important aim for ensuring food security worldwide. However, the gene regulatory network underlying phosphorus use in foxtail millet remains unclear. In this study, the molecular mechanism underlying low-phosphorus (LP) responsiveness in foxtail millet was evaluated using a comparative transcriptome analysis. LP reduced the chlorophyll content in shoots, increased the anthocyanin content in roots, and up-regulated purple acid phosphatase and phytase activities as well as antioxidant systems (CAT, POD, and SOD). Finally, 13 differentially expressed genes related to LP response were identified and verified using transcriptomic data and qRT-PCR. Two gene co-expression network modules related to phosphorus responsiveness were positively correlated with POD, CAT, and PAPs. Of these, *SiPHR1*, functionally annotated as PHOSPHATE STARVATION RESPONSE 1, was identified as an MYB transcription factor related to phosphate responsiveness. *SiPHR1* overexpression in *Arabidopsis* significantly modified the root architecture. LP stress caused cellular, physiological, and phenotypic changes in seedlings. *SiPHR1* functioned as a positive regulator by activating downstream genes related to LP tolerance. These results improve our understanding of the molecular mechanism underlying responsiveness to LP stress, thereby laying a theoretical foundation for the genetic modification and breeding of new LP-tolerant foxtail millet varieties.

## 1. Introduction

Phosphorus (P) is an essential macronutrient with an important role in plant growth and development. Two forms of soil P, organophosphate or inorganic phosphate (Pi), may be directly or indirectly taken up by plant roots [1]. Low P utilization efficiency (PUE) in soils has seriously limited crop production worldwide, leading to severe food insecurity [2]. Less than a quarter of crops use P fertilizer, and enormous losses of P via erosion and surface runoff have caused environmental degradation and contributed to water eutrophication [3]. Plants mainly adapt to variation in P availability by remodeling their root system architecture (RSA), which may involve morphological modifications, such as shortening and increasing the density of lateral roots as well as modifying the length and density of root hairs and the root-to-shoot ratio [4,5,6]. Under low-P conditions, plants scavenge and reserve internal P mainly via sugar and lipid metabolic pathways [7]. Plants are known to trigger local and long-distance P signals via hormonal secondary messengers, such as Ca^2+^, reactive oxygen species, and inositol polyphosphates [8]. Plants accumulate more auxin in their roots to alter their RSA and become more responsive to stress [9,10]. In *Arabidopsis* and rice, many key genes involved in P translocation and remobilization between the roots and shoots, such as P starvation inducible (*PSI*), phosphate transporter (*Pht*), and phosphate permease (*PHO*) gene family members, have been identified [11,12,13]. The regulatory network of the PHOSPHATE STARVATION RESPONSE 1 (PHR1)-like 1 (*PHL1*) core module, including the genes necessary for sensing Pi status and Pi starvation response (*PSR*), has been established [14]. Other transcription factors associated with Pi stress, such as *OsWRKY74* [15], *OsMYB4P* [16], *TabHLH1* [17], and *ZAT6* [18], have also been identified. Non-coding ribonucleic acids (RNAs), including miRNA399, miRNA827, and long non-coding RNAs, reportedly respond to Pi status [19,20,21]. The functions of genes involved in small ubiquitin-like modifier (SUMO)ylation, phosphorylation, dephosphorylation, and protein translocation have also been elucidated [22,23,24]. However, the complex gene regulatory networks related to P-utilization efficiency in crop plants adapted to low-phosphate (LP) soil conditions remain poorly understood.

Despite extensive research on the molecular mechanism related to the response to P stress in crops and model plants, little is known about minor crops, like foxtail millet (*Setaria italica* (L.) Beauv.). Foxtail millet, believed to have originated in China, is one of the oldest minor crops cultivated in the world [25]. It is characterized by a small genome size (~515 Mb), short life cycle, strong drought tolerance ability, and low nutrient input conditions, and has become an important C_4_ model crop for plant nutrition and molecular biology research [26]. Since publication of the genome sequences of foxtail millet by the Beijing Genomic Institute (BGI) of China and the Joint Genome Institute (JGI) of the United States Department of Energy, many studies have focused on functional genomics of the species [27,28,29]. Maps of genome availability and high-density linkage of foxtail millet provide important resources for those studying the mechanisms underlying plant nutrients, such as P use efficiency, and facilitate agricultural sustainability and food security by enhancing crop diversity [28]. A Pi deficiency in foxtail millet induced the development of a larger root system in terms of crown root length and the number, length, and density of lateral roots [30], and the use of internal Pi reserves for higher Pi utilization [31]. Previous studies have revealed that foxtail millet may undergo morphological, physiological, and biochemical changes under low phosphate stress [30,32]. However, the molecular mechanisms underlying LP stress tolerance in foxtail millet remain unclear. Therefore, in this study, we focused on the P responsiveness of foxtail millet, and identified the key genes associated with LP responsiveness to determine underlying regulatory mechanisms. Furthermore, GO (Gene Ontology), KEGG (the Kyoto Encyclopedia of Genes and Genomes) enrichment analyses, and WGCNA (Weighted Gene Co-Expression Network Analysis) were performed to elucidate key transcription factors and genes related to P responsiveness. Furthermore, the gene expression levels obtained from transcriptome data were validated via real-time quantitative reverse transcription PCR (qRT-PCR). Finally, *SiPHR1* was identified as a transcription factor and its function was validated in relation to P responsiveness. Our results are expected to fill the gap in knowledge between observed phenotypical and physiological and biochemical processes involved in low P responsiveness in foxtail millet.

## 2. Results

### 2.1. Chlorophyll and Anthocyanin Contents and Antioxidant Enzyme Activity under Phosphorus-Deficiency Treatment 

To determine the phenotypic, physiological, and biochemical characteristics of foxtail millet under LP stress, we analyzed the changes in the chlorophyll content, anthocyanin content, and antioxidant enzyme activity in the shoots and roots after LP treatment for 0, 2, 24, and 72 h. Foxtail millet seedlings exposed to LP treatment for 3 d (72 h) to 5 d exhibited a shorter and thinner root phenotype compared with the CK phenotype (Figure 1A). The chlorophyll content of leaves exposed to LP treatment for 72 h was significantly lower (by 29.13%) than that of the CK group (*p <* 0.01) (Figure 1B). The anthocyanin contents of the leaves were increased by 30.78% and 15.03% at 24 h and 72 h, respectively, compared with those of the CK group (*p <* 0.05). Furthermore, we found that the activity of phytase (Phyt) in shoots and roots exposed to LP treatment increased by 28.93% and 38.20%, respectively, at 24 h (Figure 1C). PAP activity in shoots exposed to LP treatment for 24 h was significantly increased (by 61.79%), while its activity levels in roots at 2, 24, and 72 h were increased by 22.32%, 40.71%, and 43.45%, respectively (*p <* 0.05). Superoxide dismutase (SOD), peroxidase (POD), and catalase (CAT) enzyme activities in the roots and shoots of foxtail millet exposed to LP treatment were highly induced at 2, 24, and 72 h, respectively (Figure 1D). Specifically, we found that SOD and POD activities in shoots exposed to LP treatment for 2 h were increased significantly by 11.57% (*p <* 0.01) and 67.43% (*p <* 0.05), respectively, compared with those in the CK group, and SOD, POD, and CAT activities in shoots and roots following 24 h of exposure to LP treatment were up-regulated by 174.27%, 50.65%, and 29.27%, respectively, compared with those of the CK group (*p <* 0.01). 

### 2.2. Transcriptome Analyses of Foxtail Millet Shoots and Roots Exposed to LP Treatment 

To determine the molecular mechanisms underlying P responsiveness in *S. italica*, we performed a comparative transcriptomic analysis of shoots and roots treated with NP and LP. Twelve paired-end libraries constructed from the shoots and roots of five-leaf seedlings were generated from over 200 GB of raw data, and 90% of the reads were successfully mapped to the reference genome of foxtail millet (Appendix A). A total of 11,222 unique differentially expressed genes (DEGs) were obtained by comparing NP and LP treatments. Furthermore, DEGs in six combinations involving shoots and roots, exposed to NP and LP treatments at three time points were determined (Figure 2A–F). Most DEGs were found in shoots at 2 h, (i.e., 4662 up-regulated and 333 down-regulated genes). A Venn diagram analysis revealed that there were 39 and 221 common DEGs in roots and shoots, respectively, at the three time points. There were 4574 DEGs specifically expressed in shoots at 2 h (Figure 3A). A lower number of DEGs, i.e., 259, were specifically expressed in the roots at 2 h (Figure 3B).

In a GO enrichment analysis, 3911 DEGs were assigned to biological process, cellular component, and molecular function GO categories. The majority of enriched GO terms were “cellular process”, “metabolic process”, “cell part”, “binding” or “catalytic activity” (Figure 3C). A KEGG enrichment analysis showed that these DEGs were significantly enriched in 33 pathways, and most DEGs could be mapped to “biosynthesis of secondary metabolites,” “MAPK signaling pathway”, or “plant hormone signal transduction” (Figure 3D). According to functional annotation and reference queries, 13 candidate genes were identified from DEGs at the three time points, which mainly included nicotianamine aminotransferase A, phosphate-induced protein, ferredoxin-nitrite reductase, purple acid phosphatase 29, mitochondrial phosphate carrier protein 1, TRANSPARENT TESTA GLABRA 1, and PHOSPHATE STARVATION RESPONSE 1 (Table 1). 

The expression levels of the DEGs detected via RNA-Seq data were validated via qRT-PCR analysis. We found that most gene expression patterns determined using qRT-PCR were consistent with the transcriptomic data (Figure 4A,B). The expression patterns of Si0g10340 (nicotianamine aminotransferase A) in roots exposed to LP treatment for 2, 24, and 72 h showed a gradual decrease, with the highest expression level at 2 h. However, the expression patterns of *Si1g05330*, *Si1g33510*, *Si9g36610*, and *Si4g21330* showed a gradual increase with the highest expression levels at 72 h. The expression pattern of *Si9g40880* was up-regulated at 2 and 72 h, and down-regulated at 24 h. The expression of *Si9g08330* in shoots exposed to LP treatment for 2, 24, and 72 h, was higher at 2 h and showed a gradual decrease thereafter. *Si1g05330*, *Si1g33510*, *Si2g30500*, *Si4g21330*, *Si2g29010*, and *Si9g36610* showed gradual increases, with higher expression at 72 h. The expression of *Si9g4088* was up-regulated at 2 h but down-regulated at 72 h. 

### 2.3. Gene Co-Expression Network Analysis of Foxtail Millet Related to the Pi Response

To construct gene regulatory networks under NP and LP treatments at the three time points, all DEGs of the shoots and roots were categorized into 16 and 22 modules, respectively, based on a scale-free topological model (β = 8; Figure 5A,B). To determine the correlations between the enzyme activities of SOD, POD, CAT, PAP, and Phyt activities, and DEGs, module−trait relationships were evaluated by calculating the Pearson correlation coefficients between the eigengene of each module and trait. In shoots, the MEcoral2, Melavenderblush3, and MEgrey modules were positively correlated with CAT and PAP activities (correlation coefficient > 0.5, *p* < 0.05). The ME mediumorchid module was significantly and positively correlated with POD and PAP activities. Methistle2, MEblue2, and Medarkorange2 were positively correlated with Phyt activities. In the roots, Mehoneydew was positively associated with SOD and PAP activities (correlation coefficient > 0.7, *p* < 0.01). MEblack and MEblue2 were positively correlated with Phyt (correlation coefficient >0.6, *p* < 0.01).

Furthermore, we constructed a co-expression network including the candidate genes *Si9g40880, Si3g05420*, and *Si2g29010* located in the Mecoral2 module. Another co-expression network included *Si1g27930*, *Si2g42340*, *Si9g08330*, and *Si1g33010* located in the Meblue2 module (Figure 5C,D). 

### 2.4. Gene Structure, Protein Domains, and Analysis of SiPHR1 in Relation to LP Responsiveness

A BLAST analysis demonstrated that the protein sequence of Si9g40880 in foxtail millet showed a 48.30% similarity with PHOSPHATE STARVATION RESPONSE (*AtPHR1,* At4G26180.1) in *Arabidopsis*. Hence, we named *Si9g40880,* functional annotation as an MYB-domain transcription factor *SiPHR1*. These gene may be related to Pi responsiveness in foxtail millet. The protein homology levels between *SiPHR1* of millet and other species were *S. viridis* (100%), *S. bicolor* (80.54%), *O. sativa* (67.05%), *Z. mays* (79.73%), and *T. aestivum* (76.45%). A phylogenetic tree of the PHR1 homologous in these species was constructed using the neighbor-joining (NJ) method. Proteins were classified into two groups. This analysis suggested that PHR1 was evolutionarily conserved among Gramineae crops (Figure 6A).

The predicted theoretical isoelectric point value, relative molecular mass, instability coefficient, and mean hydrophobicity index of *SiPHR1* were 6.02, 34,223.66 Da, 46.47 (>40 is unstable), and −0.0886, respectively. The tertiary protein structure of *SiPHR1* consisted of 30.35% α-helical structure, 59.74% irregular coils, 3.19% folding, and 6.71% extended chains (Figure 6B). A gene structure analysis showed that the open reading frame of *SiPHR1* was 1344 bp, encoding 448 amino acid residues with two conserved domains (SANT superfamily and MYB-CC type transfactor) located on chromosome 9 (Figure 6C). 

A cis-acting element analysis was performed for the 2000 bp upstream promoter sequence of *SiPHR1* using the plant cis-acting regulatory elements (PlantCARE) tool (Figure 6D). In addition to the core elements of the promoter (TATA-box and CAAT-box), cis-acting elements related to abiotic stress, hormone signaling, and growth development regulation, such as photo-responsive elements (G-Box/T-Box/CT-motif/GT1-motif/metal response element (MRE)/Box4/Sp1), abscisic-acid-responsive (ABRE), methyl jasmonate (MeJA)-responsive (CGTCA-motif/TGACG-motif), gibberellin-responsive (P-box), and MYB binding site element (CCAAT-box), were detected.

### 2.5. Subcellular Location and SiPHR1 Overexpression in Arabidopsis enhanced LP Tolerance Ability

The empty vector *p35::mGFP* and *SiPHR1* fused with the modified green fluorescent protein (mGFP) vector *p35::SiPHR1-mGFP* were transferred into *A. tumefaciens* LBA4404. These strains were then injected into *Nicotiana. tabacum* leaves and grown in the dark for 2 d. Confocal laser microscopy indicated that *p35::SiPHR1-mGFP* emitted green fluorescence under ultraviolet (UV) excitation and red fluorescence as a nuclear localization signal. The merged signal represented by yellow fluorescence showed that SiPHR1 was located in the nucleus. 

Two transgenic plants (TPs), namely *Ubi::SiPHR1-GFP-8* and *Ubi::SiPHR1-GFP-9*, which showed higher expression levels than those in control plants and were used to further analyze phenotypes under LP treatment. The root morphology of TP in the LP-treated Murashige and Skoog (MS) medium was altered significantly, as evidenced by more lateral roots (Figure 7A) and root hairs than those in wild type (WT) plants (Figure 7B). Exposure to LP stress for 7 d inhibited primary root elongation in TP and WT plants. Compared with LP, the main root length of WT decreased by 43.91% and those of Ubi:SiPHR1-8 and Ubi:SiPHR1-9 of TP decreased by 14.42% and 22.7%, respectively. However, the length of root hairs in TP and WT plants was significantly increased under LP treatment. The lengths of root hairs in the WT and two TP plants increased by 193.84%, 63.21%, and 21.19%, respectively. The number of lateral roots in TP was significantly higher than that in WT under both LP and NP treatments (*p* < 0.01). Lengths of *Ubi::SiPHR1-GFP-8* in TP plants were significantly higher than those of WT under LP and NP treatments. Considered together, these results suggest that *SiPHR1* is associated with LP responsiveness in foxtail millet and may alter the root morphology of seedlings to manage LP stress.

## 3. Discussion 

Reactive oxygen species (ROS) play crucial roles in abiotic and biotic stress-sensing and thereby affect plant development [33]. ROS production is eliminated mainly by antioxidant enzyme systems, such as SOD, POD, and CAT, as well as non-enzymatic systems, such as glutathione, ascorbate, and carotenoids [34,35]. Deficiencies in nitrogen (N), phosphorus (P), and potassium (K) may induce ROS production and enhance antioxidant enzyme activity in *Arabidopsis* and other crops [36,37]. In rice, a phosphorus deficiency did not alter photosystem II (PSII) photochemistry efficiency but increased the activities of SOD, POD, and CAT in roots, suggesting that photoprotective mechanisms contribute to the response to LP stress [38]. Previously, we have reported that antioxidant enzyme activity may be increased under LP stress to eliminate H_2_O_2_ from the root systems of different foxtail millet genotypes [39]. We also found similar results in this study, i.e., SOD, POD, and CAT enzyme activities in shoots and roots increased 0.2~1.7-fold at 24 h under LP stress in foxtail millet. These results suggested that the peroxidation process and enzyme-protection system of plant cells are enhanced by increasing SOD, POD, and CAT enzyme activities under LP stress. In maize, the chlorophyll content of the LP-treatment mutant was significantly higher than that of the wild type (0.5-fold), and a P deficiency resulted in a decrease in the chlorophyll content [40]. Elevating anthocyanin levels could alleviate growth inhibition, yield loss, and oxidative destruction accumulation in maize [41]. We also found that LP stress decreased the chlorophyll content at 72 h in the LP-sensitive genotype of foxtail millet. Anthocyanin accumulation was also observed under LP stress at 24 and 72 h in foxtail millet. These findings suggest that the metabolism of flavonoids and chlorophyll may be affected by ROS production. 

Plant PAPs catalyze the hydrolysis of phosphate esters and anhydrides in acidic environments and facilitate diverse physiological functions, including phosphorus acquisition, ROS production, flower development, and cell-wall biosynthesis [42]. Many plant PAPs induced by P deficiency belong to the metallophosphoesterase superfamily, suggesting that they may play a role in P acquisition and assimilation [43]. A higher than four-fold increase in acid phosphatase activity was induced by phosphorus starvation in the leaves of the phosphorus-inefficient common bean variety, DOR364 [42]. Association mapping of seed traits and phytate content combined with recombinant enzyme activity analysis revealed the potential role played by *CaPAP7* in seed phytate accumulation in chickpeas [43]. PAPs also utilize a wide range of substrates involved in phytate hydrolysis to release Pi for use in plant cells or rhizospheres [44]. Our results showed that the acid phosphatase and phytase activities in the shoots were increased 0.2~0.4-fold by LP stress. This suggests that foxtail millet seedlings exhibit increases in effective internal phosphorus contents and underlying phosphorus utilization efficiency by enhancing the activity of acid phosphatase and phytase. Overall, our study revealed that LP tolerance in foxtail millet may be increased by coordinating the degradation of chlorophyll, synthesis of anthocyanins, antioxidant enzymes, acid phosphatase, and phytase via key regulatory genes and related physiological and biochemical pathways. 

Many other key genes induced by P deficiency have been identified via transcription profiling and gene-expression changes. For example, transcriptomic profiling of Pi-starved barley shoots identified 98 DEGs involved in plant defense, plant stress response, nutrient mobilization, and phosphorus metabolism according to GO analyses [45]. In total, 5900 and 3389 DEGs were identified in low-P-sensitive 31778 and low-P-tolerant CCM454 maize varieties, respectively. GO enrichment analyses revealed that these DEGs were involved in various metabolic processes, including PAP activity [46]. A total of 2128 DEGs, showing changes of 2-fold or greater (*p* ≤ 0.05), were identified in white lupin in response to Pi deficiency, while 12 candidate DEGs consistently related to the Pi status of plants have been identified in *Arabidopsis,* potato, and white lupin [47]. Based on an RNA-seq analysis, we identified 39 and 221 common DEGs in the roots and shoots, respectively, between LP and NP, at the three time points. A GO analysis revealed 3911 DEGs enriched in the terms “cellular process”, “metabolic process”, and “catalytic activity” (Figure 3C). We detected fewer DEGs in the root and shoot of foxtail millet than in other crops in previous studies. Furthermore, most DEGs were involved in the “biosynthesis of secondary metabolites,” “MAPK signaling pathway”, and “plant hormone signal transduction” KEGG pathways (Figure 3D). These results showed that a few key genes related to hormone, metabolism, and signal-transduction pathways in plants responded to LP stress. A complex regulatory network was related to phosphorus metabolism and LP responsiveness.

In maize, two prominent co-expression modules obtained by WGCNA, which included proteins and genes related to sucrose biosynthesis, lipid metabolism, respiration, and RNA processing, were found to be positively correlated with root and shoot traits in the maize phosphate-starvation response [48]. Based on WGCNA analysis, we found that two co-expression network modules, MEcoal2 and MEblue, were positively correlated with root and shoot traits under LP stress. Two co-expression networks were constructed using the following seven candidate hub genes: *PHR, PAP29, SIZ1*, *TRANSPARENT TESTA GLABRA 1, Potassium transporter 7, zinc finger protein 1,* and *phosphate-induced protein 1*. These hub genes related to LP responsiveness have also been identified in *Arabidopsis*, maize, and other crops [14,18,49,50,51]. Our results suggest that PHR, TRANSPARENT TESTA GLABRA 1, and zinc finger protein 1 are potential key regulators of genes related to P responsiveness. A previous study has reported that PHR1 is sumoylated by SIZ1, which promotes its activities via post-translational modifications under LP conditions [52]. Another study has shown that PHR1 directly binds to the PIBS motifs on the promoter of F3′ H and LDOX and up-regulates their expression, resulting in anthocyanin accumulation [53]. In this study, we identified the sequence characteristics of *SiPHR1*, which encodes a type of MYB-cc DNA-binding domain. These results suggested that SiPHR1 binds to downstream genes related to P responsiveness. Specifically, we hypothesized that PHR may directly or indirectly regulate PAP29 expression under LP conditions. A previous study has reported that PHR1 is sumoylated by SIZ1, which promotes its activities via post-translational modifications under LP conditions.

Further, under LP treatment, SiPHR1 overexpression altered the root structure and morphology, root length, root number, and root hair length. SiPHR1 alleviated P stress and promoted P transport and uptake in plants. Overall, these results suggest that SiPHR1, may be sumoylated by SIZ1 to regulate downstream genes, such as PAP29, under LP stress. However, the regulatory network related to LP responsiveness in plants appears to be complex. Our study provides a better understanding of the molecular mechanism(s) underlying LP responsiveness in foxtail millet, a plant with a small root system that copes with poor soil conditions.

## 4. Materials and Methods

### 4.1. Plant Materials

The B027 variety of foxtail millet, a Pi-sensitive cultivar (unpublished research), was donated by the Center for Crop Germplasm Resources of the Chinese Academy of Agricultural Sciences(Beijing, China). The donated seeds were planted and grown in a stable greenhouse environment at Shanxi Agricultural University(Jinzhong, China). The seeds were sterilized using 1% NaClO solution and immersed in water at 25 °C for 2 d followed by germination at 28 °C with a photoperiod of 16 h (light)/8 h (dark) and a relative humidity of 70%. Hydroponic seedlings of B027 were grown for three weeks in 2000 mL boxes containing 1/2 strength Hoagland nutrient solution. Seedlings at the five-leaf stage were then transferred to Hoagland solution with 1 mM KH_2_PO_4_ (normal Pi treatment, NP) or 5 µM KH_2_PO_4_ (low Pi treatment, LP), according to a previous study [54]. Shoot and root tissues of B027 were collected at 2, 24, and 72 h in LP and NP (CK) for further analyses. Each treatment consisted of three replicates. All samples were immersed in liquid nitrogen immediately after collection and stored at –80 °C for further analyses. 

### 4.2. Determination Content of Chlorophyll and Anthocyanin Contents

Fresh leaves and roots of foxtail millet seedlings were collected and weighed to determine their chlorophyll and anthocyanin contents. Chlorophyll and anthocyanins were extracted from ground samples (0.1 g) using the acetone and methanol-HCl extraction methods described in previous studies. [55,56]. The light-absorption values of each extracted sample, at 645, 663, and 652 nm for chlorophyll, and at 530 and 657 nm for anthocyanin, were detected via spectrophotometry(Shanghaijingke Co., Shanghai, China). Three biological replicates were used for all samples and determinations. 

### 4.3. Determination of Antioxidant Enzyme, Purple Acid Phosphatase, and Phytase Activities

In order to measure enzymatic activities, root and shoot samples (approximately 0.5 g) were crushed in liquid nitrogen and extracted using 10 mL of a 50 mM sodium phosphate buffer. The extracted solution was centrifuged at 12,000× *g* for 10 min at 4 °C. Finally, the collected supernatant was stored at 4 °C, and the SOD, POD, and CAT activities were measured according to the standard method described in the literature [57]. 

PAP and phytase activities were detected using an enzyme-linked immunosorbent assay (ELISA) kit (Thermo Fisher Scientific Inc.,Waltham, USA; COIBO BIO Co.; Shanghai, China) according to the manufacturer’s instructions. Briefly, powdered leaf samples were extracted using ice-cold extraction buffer (100 mM potassium acetate, pH 5.5, 20 mM CaCl_2_, 2 mM ethylenediaminetetraacetic acid (EDTA), 1 mM dithiothreitol (DTT), 0.1 mm phenylmethylsulphonyl fluoride, and 1.5% (*w*/*v*) polyvinylpolypyrrolidone). Then, the supernatant was obtained by centrifugation at 15,000× *g* for 15 min at 4 °C. Next, 50 μL of the supernatant were added to a 96-well plate with 100 μL of detection antibody and allowed to react in the water bath at 37 °C for 60 min; 50 μL of substrates was added to the 96-well plate and allowed to react in the water bath at 37 °C for 15 min. The liquid was then discarded and samples were washed. Finally, the reaction was stopped and the optical density was detected at 450 nm. 

### 4.4. RNA Extraction and RNA-seq

Total RNA was isolated from frozen shoot and root tissues using a TRIzol kit (Invitrogen, Carlsbad, CA, USA) according to the manufacturer’s instructions. The cDNA library was constructed and sequenced using the Illumina HiSeq 2100 platform (Illumina Inc., San Diego, CA, USA), with three biological replicates for each sample. 

### 4.5. Analysis of RNA-seq Data and Differentially Expressed Genes

After removing the adapters and low-quality reads, the clean reads were mapped to the foxtail millet reference genome, *S. italica* v2.2 using hierarchical indexing for spliced alignment of transcripts 2 (Hisat2) version 2.2.1.0 (http://phytozome.jgi.doe.gov/, accessed on 10 march 2021) [58]. All transcripts were assembled using StingTie (ccb.jhu.edu/software.shtml accessed on 15 march 2021) [59]. Gene expression values were calculated based on fragments per kilobase of exon model per million mapped fragments (FPKM) using Cufflinks version 2.2.1 software (cole-reapnell-lab.github.io/cufflinks/tools/, accessed on 1 April 2021) [60]; DEGs were identified using DEseq version 2 with the R package functions, ‘estimateSizeFactors’, and ‘nbinom Test’ [61]. The screening thresholds for DEGs was |FC (fold change)| > 0.5 and false discovery rate (*p*-value) <0.05. Venn and UpSet diagrams were drawn using Toolkit for Biologists (TB) tools with default settings [62]. 

### 4.6. Functional Annotation and WGCNA Analysis of DEGs

DEGs associated with low P responsiveness were identified by comparing different datasets using a Venn diagram analysis and functional annotation via non-redundant (NR), GO, Clusters of Orthologous Genes (COG), and KEGG databases. GO and KEGG enrichment analyses were performed using ClusterProfiler [63]. A heatmap was generated using the “heat map” R-package. To identify key genes highly associated with low-P responsiveness in foxtail millet, gene co-expression networks were constructed using the WGCNA package in R [45]. Core DEGs were further divided into modules, following which the correlation between each module and LP responsiveness was calculated. Module–trait associations were estimated using the correlation between the module eigengene and enzyme activity data under low Pi/control treatments. Module identification was implemented after merging modules with expression profiles like a merge-cut height of 0.5. Interaction network visualization for each module was performed using Cytoscape version 3.8.0 [64]. The gene co-expression network is a scale-free weighted gene network with multiple nodes connected to different nodes via edges. Each node represents a gene connected to a different number of genes by weight.

### 4.7. qRT-PCR Analyses

The expression levels of 13 DEGs were detected using qRT-PCR. Total RNA was extracted using TRIzol reagent according to the manufacturer’s protocol (TaKaRa, Dalian, Liaoning, China). The primer-BLAST online National Center for Biotechnology Information (NCBI) tool was used to design primers specific for each DEG (Appendix A). The cDNA was synthesized using a RevertAid First Strand cDNA Synthesis Kit (TaKaRa), and qRT-PCR was performed following the protocol for SYBR Green PCR Master Mix Reagent (SuperReal PreMix Plus, Tiangen, China) on a Bio-Rad Real-Time PCR System (Carlsbad, CA, USA). Actin was used as the internal control and the relative expression level of each assessed gene was calculated by using the method 2^−ΔΔCt^ method [65].

### 4.8. Vector Construction and Genetic Transformation of Arabidopsis and Tobacco

Total RNA was extracted from the leaves of variety B27, and cDNA was obtained using RT-PCR. The coding sequence of SiPHR1 was amplified using gene-specific forward and reverse primers (Appendix A). The coding sequence (CDS) of *SiPHR1* was constructed into *pCAMBIA1302-mGFP* and pCAMBIA1300 and transformed into *E. coli* DH5α cells. The overexpression vectors *p35S::SiPHR1-mGFP* and *Ubi::SiPHR1-GFP* were transformed into *A. tumefaciens* GV3101.

Using the *Arabidopsis* floral-dip method, the *A. tumefaciens* GV3101 strain carrying the *Ubi:: SiPHR1-GFP* vector was genetically transformed into WT *Arabidopsis*. Gene-specific PCR detection and hygromycin resistance screening was used to obtain T3 generation TPs for phenotypic and gene expression analysis. For subcellular localization analysis, *A. tumefaciens* GV3101 containing the *35S::SiPHR1-mGFP* vector was injected into the leaves of 30 d *N. benthamiana* seedlings using the transient transformation method. The infected leaves were grown in a dark environment for 3 d and then observed under a laser confocal microscope.

## 5. Conclusions

The low-phosphorus stress reduced the chlorophyll content in shoots, but increased the anthocyanin content in the roots in foxtail millet. The activities of PAP, phytase, CAT, POD, and SOD were also up-regulated induced by LP treatment. Some differentially expressed genes were identified and verified by using transcriptomic data and qRT-PCR. Two gene co-expression networks correlated with POD, CAT, and PAPs were constructed by WGCNA. *SiPHR1* was one of the key gene involved in the low-phosphorus-responsive network, verified as an MYB transcription factor, it could modify the root architecture by increasing the main root length and the numbers of lateral roots and root hairs. The findings of this study shed light on the response mechanism of foxtail millet to low-phosphorus stress and lay a theoretical foundation for the genetic modification and breeding of low-phosphorus-tolerant foxtail millet.

## Figures and Tables

**Figure 1 ijms-24-12786-f001:**
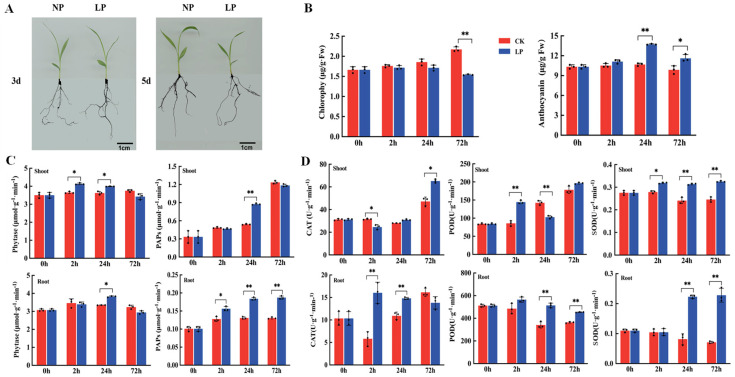
Comparison of physiological and biochemical indexes of foxtail millet at three time points under low Pi treatment. (**A**) Phenotype analysis of the B27 variety under low phosphate (LP) and normal phosphate (NP) treatment; (**B**) the chlorophyll and anthocyanin contents of B27 under LP and NP treatment; (**C**) activities of PAP and phytate in the shoot and root under LP and NP treatment; and (**D**) activities of SOD, CAT, and POD in the shoot and root under LP and NP treatment. Note: * and ** indicated significant difference at *p* < 0.05 and 0.01 level using paired *t*-test approach.

**Figure 2 ijms-24-12786-f002:**
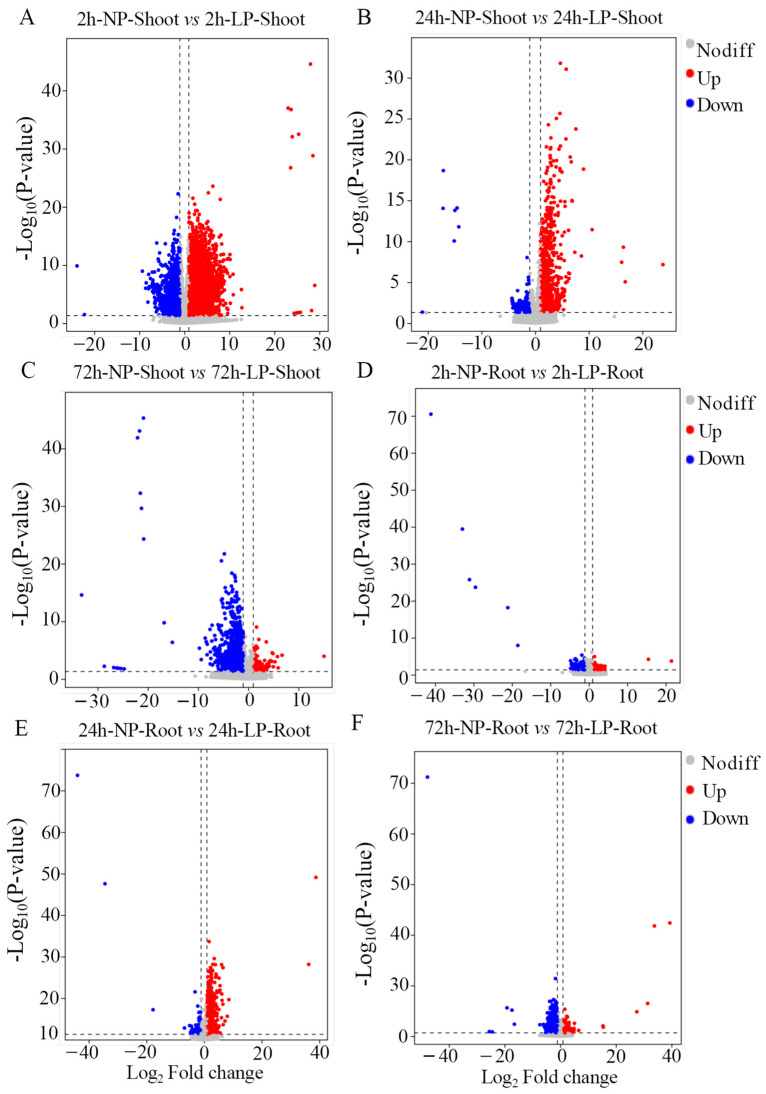
Analysis of DEGs between LP and NP treatments in the shoot and root at three time points. (**A**) 2 h NP vs. LP in the shoot; (**B**) 24 h NP vs. LP in the shoot; (**C**) 72 h NP vs. LP in the shoot; (**D**) 2 h NP vs. LP in the root; (**E**) 24 h NP vs. LP in the root; and (**F**) 72 h NP vs. LP in the root.

**Figure 3 ijms-24-12786-f003:**
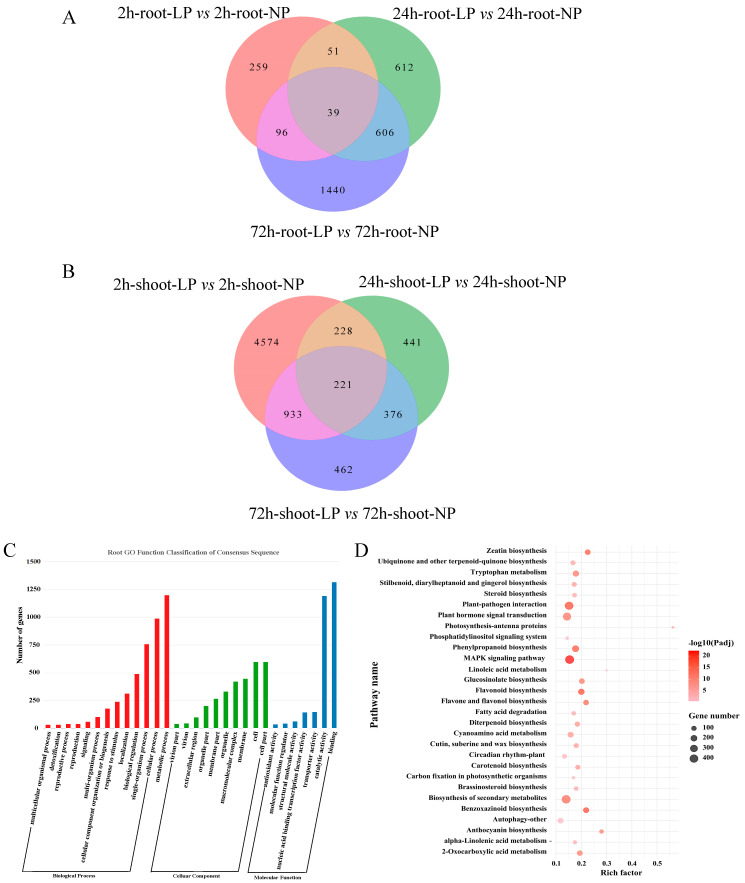
Venn diagram and GO and KEGG pathway enrichment analyses of DEGs based on comparative transcriptomic data. (**A**) Venn diagram of DEGs in the root for LP vs. NP at three time points; (**B**) Venn diagram of DEGs in the shoot for LP vs. NP at three time points; (**C**) GO term enrichment analysis of all DEGs derived from the root and shoot of LP vs. NP at three time points; and (**D**) KEGG pathway enrichment analysis of all DEGs derived from the root and shoot of LP vs. NP at three time points.

**Figure 4 ijms-24-12786-f004:**
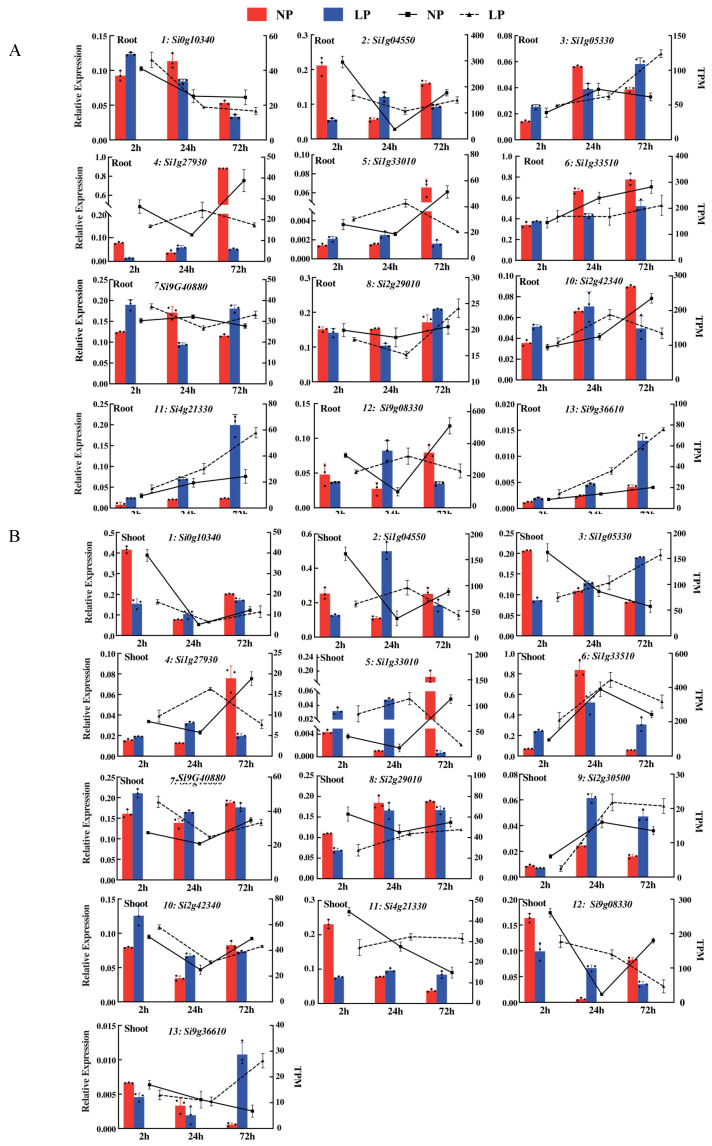
Expression patterns of 13 candidate genes in the root and shoot after LP and NP treatments at three time points. (**A**) FPKM values and gene expression levels determined using qRT-PCR of candidate genes in the root and (**B**) FPKM values and gene expression levels determined using qRT-PCR of candidate genes in the shoot.

**Figure 5 ijms-24-12786-f005:**
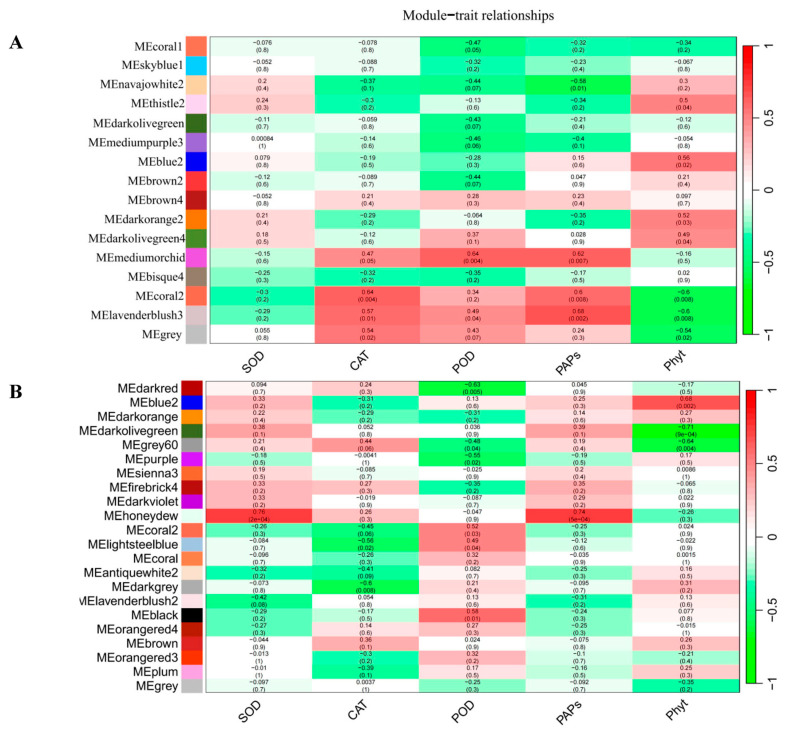
Gene co-expression network and module−trait relationship analyses. (**A**) Module−trait relationship analysis of the shoot; (**B**) module−trait relationship analysis of the root; (**C**) gene co-expression network in Mecroal2; and (**D**) gene co-expression network in Meblue2.

**Figure 6 ijms-24-12786-f006:**
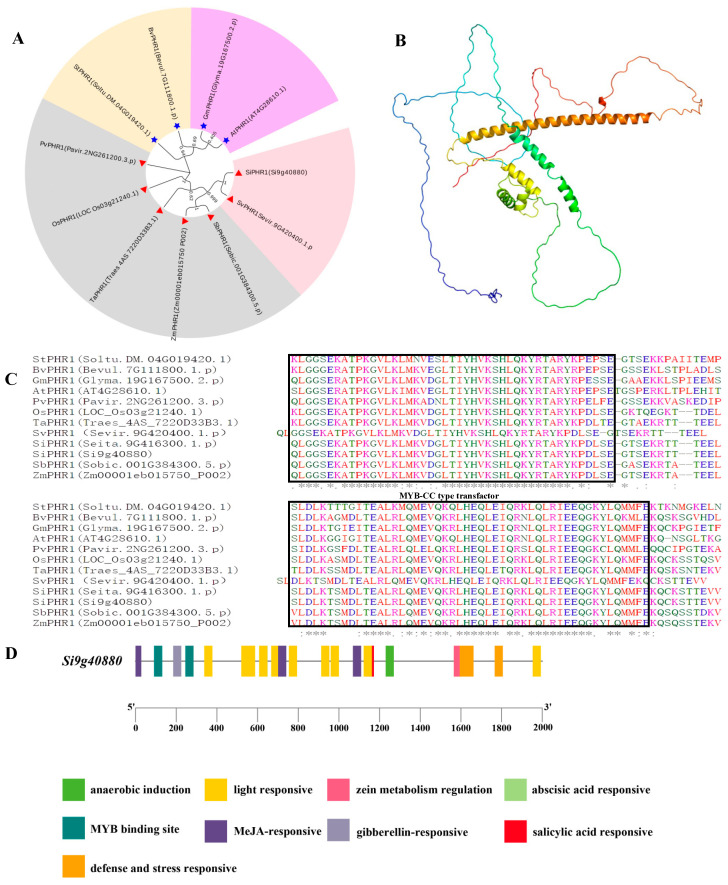
Phylogenetic tree, protein domain, sequence alignment, and cis-acting element analysis of *SiPHR1*. (**A**): Phylogenetic tree of PHR1 derived from *A. thaliana*, *Z. mays*, *S. bicolor*, *S. virira*, *G. max*, *P. virgatum*, *B. vulg*, and *S. tuber*; (**B**): protein structural domains; (**C**): sequence characteristic of the conserved domain of *SiPHR1* and other species; * represents conserved amino acid residues among these species. and (**D**): cis-regulatory element analysis of *SiPHR1*(Si9g40880). Note: different background color indicated different groups in phylogenetic tree; blue five-pointed star indicated these genes belong to dicotyledon; red triangle indicated these genes belong to monocotyledon.

**Figure 7 ijms-24-12786-f007:**
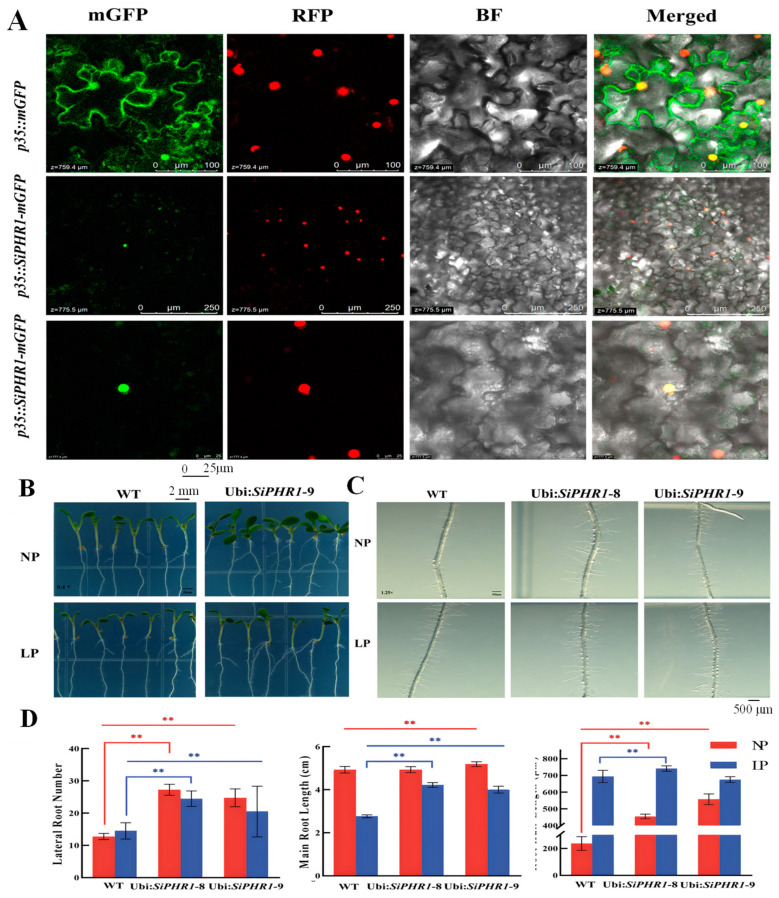
Analysis of root phenotypes in *SiPHR1*-overexpressing transgenic plants under LP stress. (**A**,**B**). Phenotype of *SiPHR1*-overexpressing TP under LP and NP treatments. (**C**,**D**) Statistical analysis of the number of later roots, the length of main root, and length of root hairs. ** *p* < 0.01 level by *t*-test analysis.

**Table 1 ijms-24-12786-t001:** Gene length, expression level, and functional annotation of the candidate genes related to LP responsiveness.

Gene ID	Gene Length (bp)	Gene Expression Level (FC)	Comparison Combination (NP vs. LP at Three Time Points)	Functional Annotation (NR/Homologous Genes in *Arabidopsis*)
Si0g10340	1209	−2.04, −1.26	A	Nicotianamine aminotransferase A
Si1g04550	516	−3.14, −1.65, 1.39	A, B, E	Uncharacterized protein
Si1g05330	849	2.58, 1.36, −1.00, 1.13, 0.15	A, C, D, E, F	SPX domain-containing protein 1
Si1g27930	1080	−0.04, 1.48, 1.36, −1.18	B, E, D, F	TRANSPARENT TESTA GLABRA 1 (AT5G24520)
Si1g33010	939	6.7, 1.74, 2.74, −2.65, −1.66	A, B, C, D, E	Phosphate-induced protein 1 (AT4G08950)
Si1g33510	1782	1.09	A	Ferredoxin-nitrite reductase (AT2G15620)
Si9g40880	1347	−0.65, −0.52	A, D	PHOSPHATE STARVATION RESPONSE 1 (AT4G28610)
Si2g29010	1200	−1.23	A	Purple acid phosphatase 29 (AT5G63140)
Si2g30500	954	−2.49, −1.31	A, D	Phosphate carrier protein (AT2G17270)
Si2g42340	2355	−1.30, 1.28	B, C	Potassium transporter 7 (AT2G30070)
Si4g21330	912	2.33, 1.21, 0.69	C, D, E	SPX domain-containing protein 1 (AT5G20150)
Si9g08330	1548	1.74, −2.00, 2.56, −0.56	B, C, D, E	Zinc finger protein 1 (AT5G04340)
Si9g36610	759	1.67, 2.21, 0.75	C, D, E	SPX domain-containing protein 5 (AT2G45130)

Note: FC indicates fold change for the comparison with *p* < 0.01. A: 2 h NP vs. LP in the shoot; B: 24 h NP vs. LP in the shoot; C: 72 h NP vs. LP in the shoot; D: 2 h NP vs. LP in the root; E: 24 h NP vs. LP in the root; F: 72 h NP vs. LP in the root.

## Data Availability

All datasets supporting the results of this study are included within the article and the Appendix A.

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
