# Peer review of "Role of SiPHR1 in the Response to Low Phosphate in Foxtail Millet via Comparative Transcriptomic and Co-Expression Network Analyses"

_ijms, 2023, doi:10.3390/ijms241612786_

Round 1

Reviewer 1 Report

To,

The Chief Editor,

IJMS, MDPI,

Manuscript ID: ijms-2469279

Subject: Submission of comments of the manuscript in “IJMS"

Dear Chief Editor IJMS, MDPI,

Thank you very much for the invitation to consider a potential reviewer for the manuscript (ID: ijms-2469279). My comments responses are furnished below as per each reviewer’s comments. 

In the reviewed manuscript, the authors uncover the molecular mechanism underlying low-phosphorus responsiveness in foxtail millet by identifying candidate genes via comparative transcriptome analysis. Results: the low-phosphorus treatment, reduced the chlorophyll content in shoots, but increases the anthocyanin content in the roots. Purple acid phosphatase and phytase activities in shoots as well as roots were upregulated at 24h. Antioxidant systems (CAT, POD, and SOD) in the shoots and roots were significantly upregulated at 72h. Finally, 13 differentially expression genes related to LP response were identified and verified using transcriptomic data and qRT-PCR. Two gene co-expression network modules related to phosphorus responsiveness were constructed using WGCNA analysis and were positively correlated with POD, CAT, and PAPs. Of these, SiPHR1, functionally annotated as PHOSPHATE STARVATION RESPONSE 1, was used for genetic transformation and functional verification as a transcription factor. Subcellular location, phylogenetic tree, and protein domain analysis showed that SiPHR1 is a type of MYB transcription factor related to phosphate responsiveness. Overexpression of SiPHR1 in Arabidopsis significantly modified the root architecture by increasing the main root length and the number of lateral roots and root hairs. Thus, low-phosphorus stress causes cellular as well as physiological and phenotypic changes in foxtail millet seedlings. Downstream genes related to low-phosphorus responsiveness activated SiPHR1, a positive regulator, which improved the capacity of foxtail millet to tolerate low-phosphorus. Our results may enable those concerned to gain a better understanding of the molecular mechanism underlying foxtail millet responsiveness to low-phosphorus stress, thereby laying a theoretical foundation for the genetic modification and breeding of new low-phosphorus-tolerant foxtail millet varieties. There are some minor and major comments. The author should revise as per my comments and suggestions. 

  1. I have read the entire manuscript and my initial comment is that manuscript is poorly written. I have significant concerns about the grammar and vocabulary of the manuscript; therefore, I recommend the authors to use an English proofreading service.
  2. The abstract does not reflect the whole story, revise it
  3. The key words must be in alphabetical order.
  4. The writing style of the paper is very poor. There are many grammatical mistakes. Long sentences with noticeable grammatical mistakes are frequently present throughout the manuscript. There are many typos mistakes in this whole manuscript. The author should check the whole manuscript.
  5. The introduction part is not impressive and systematic. In the introduction part, the authors should elaborate on the scientific issues in plant research. The Content of the introduction is effective in essence but very poorly presented, significant improvements are needed in presenting the proper background of the work undertaken
  6. The figures are quite low resolution and difficult to make out. Higher-resolution versions will be needed for publication. Further, text in figure is not readable, for example, in Figures 1, 2, 3, 4, 5, 6 and 7.
  7. The problem is not with the research, but with the form of the description, which is too poor in my opinion. The literature review and discussion should be improved. The discussion should be interpreted with the results as well as discussed in relation to the present literature. 
  8. References: shall have to correct the whole References according to the ”Instructions for the Authors”, e.g. the Journal name must be abbreviated, journal name in italics, the year must be bold and you shall have to use the abbreviated number of the Journals cited.
  9. Authors must add the conclusion section.

To,

The Chief Editor,

IJMS, MDPI,

Manuscript ID: ijms-2469279

Subject: Submission of comments of the manuscript in “IJMS"

Dear Chief Editor IJMS, MDPI,

Thank you very much for the invitation to consider a potential reviewer for the manuscript (ID: ijms-2469279). My comments responses are furnished below as per each reviewer’s comments. 

In the reviewed manuscript, the authors uncover the molecular mechanism underlying low-phosphorus responsiveness in foxtail millet by identifying candidate genes via comparative transcriptome analysis. Results: the low-phosphorus treatment, reduced the chlorophyll content in shoots, but increases the anthocyanin content in the roots. Purple acid phosphatase and phytase activities in shoots as well as roots were upregulated at 24h. Antioxidant systems (CAT, POD, and SOD) in the shoots and roots were significantly upregulated at 72h. Finally, 13 differentially expression genes related to LP response were identified and verified using transcriptomic data and qRT-PCR. Two gene co-expression network modules related to phosphorus responsiveness were constructed using WGCNA analysis and were positively correlated with POD, CAT, and PAPs. Of these, SiPHR1, functionally annotated as PHOSPHATE STARVATION RESPONSE 1, was used for genetic transformation and functional verification as a transcription factor. Subcellular location, phylogenetic tree, and protein domain analysis showed that SiPHR1 is a type of MYB transcription factor related to phosphate responsiveness. Overexpression of SiPHR1 in Arabidopsis significantly modified the root architecture by increasing the main root length and the number of lateral roots and root hairs. Thus, low-phosphorus stress causes cellular as well as physiological and phenotypic changes in foxtail millet seedlings. Downstream genes related to low-phosphorus responsiveness activated SiPHR1, a positive regulator, which improved the capacity of foxtail millet to tolerate low-phosphorus. Our results may enable those concerned to gain a better understanding of the molecular mechanism underlying foxtail millet responsiveness to low-phosphorus stress, thereby laying a theoretical foundation for the genetic modification and breeding of new low-phosphorus-tolerant foxtail millet varieties. There are some minor and major comments. The author should revise as per my comments and suggestions. 

  1. I have read the entire manuscript and my initial comment is that manuscript is poorly written. I have significant concerns about the grammar and vocabulary of the manuscript; therefore, I recommend the authors to use an English proofreading service.
  2. The abstract does not reflect the whole story, revise it
  3. The key words must be in alphabetical order.
  4. The writing style of the paper is very poor. There are many grammatical mistakes. Long sentences with noticeable grammatical mistakes are frequently present throughout the manuscript. There are many typos mistakes in this whole manuscript. The author should check the whole manuscript.
  5. The introduction part is not impressive and systematic. In the introduction part, the authors should elaborate on the scientific issues in plant research. The Content of the introduction is effective in essence but very poorly presented, significant improvements are needed in presenting the proper background of the work undertaken
  6. The figures are quite low resolution and difficult to make out. Higher-resolution versions will be needed for publication. Further, text in figure is not readable, for example, in Figures 1, 2, 3, 4, 5, 6 and 7.
  7. The problem is not with the research, but with the form of the description, which is too poor in my opinion. The literature review and discussion should be improved. The discussion should be interpreted with the results as well as discussed in relation to the present literature. 
  8. References: shall have to correct the whole References according to the ”Instructions for the Authors”, e.g. the Journal name must be abbreviated, journal name in italics, the year must be bold and you shall have to use the abbreviated number of the Journals cited.
  9. Authors must add the conclusion section.

Author Response

Dear Editors and Reviewer:

   Thank you for your letter and for the reviewers’ comments concerning our manuscript entitled “ The functional characteristic of SiPHR1 as a hub gene responded for low phosphate via comparative transcriptomic and the co-expression network analysis in foxtail millet ”(ID: IJMS 2469279). Those comments are all valuable and very helpful for revising and improving our paper, as well as the important guiding significance to our researches. We have studied comments carefully and have made correction which we hope meet with approval. Revised portion are marked in color in the paper. The main corrections in the paper and the responds to the reviewer’s comments are as following:

Reviewer 1

  1. I have read the entire manuscript and my initial comment is that manuscript is poorly written. I have significant concerns about the grammar and vocabulary of the manuscript; therefore, I recommend the authors to use an English proofreading service.

Response: We are very sorry for our incorrect writing in manuscript about the grammar and vocabulary, according to your advice, we use the Editage, an English proofreading service (http://www.editage.cn) to proof the article.

  1. The abstract does not reflect the whole story, revise it.

Response: Considering the reviewer’s suggestion, we have revised the abstract, and added the experiment protocol and the purpose of the article (Line 12-20)

  1. The key words must be in alphabetical order.

Response: We are very sorry for our negligence of order about the key words, and them are modified by alphabetical order. (Line 36)

  1. The writing style of the paper is very poor. There are many grammatical mistakes. Long sentences with noticeable grammatical mistakes are frequently present throughout the manuscript. There are many typos mistakes in this whole manuscript. The author should check the whole manuscript.

Response : We are very sorry for our poor writing style and grammatical mistakes in this paper. We have checked the whole manuscript carefully.

  1. The introduction part is not impressive and systematic. In the introduction part, the authors should elaborate on the scientific issues in plant research. The Content of the introduction is effective in essence but very poorly presented, significant improvements are needed in presenting the proper background of the work undertaken

Response: It is really true as reviewer suggested that the introduction is not impressive and systematic. We have made correction according to the reviewer’s comments, and elaborated on the scientific issues about plant research, and reorganized the content of the introduction, especially about the background of the work undertaken. (1. Abstract)

  1. The figures are quite low resolution and difficult to make out. Higher-resolution versions will be needed for publication. Further, text in figure is not readable, for example, in Figures 1, 2, 3, 4, 5, 6 and 7.

Response: We have re-drawn the figures according to the Reviewer’s suggestion. (Figures 1-7)

  1. The problem is not with the research, but with the form of the description, which is too poor in my opinion. The literature review and discussion should be improved. The discussion should be interpreted with the results as well as discussed in relation to the present literature. 

Response : Thank you very much for your kindly suggestion, we have re-written the discussion part, and discussed the results as well as the relation to the present literature such as in maize. (3. Discussion, Line 296-298; 300-307; 342-343; 346-349; 359-360; 362-368; 373-375 )

  1. References: shall have to correct the whole References according to the ”Instructions for the Authors”, e.g. the Journal name must be abbreviated, journal name in italics, the year must be bold and you shall have to use the abbreviated number of the Journals cited.

Response: We have made correction about the whole references according to the “instructions for the Authors”, and correct it as the IJMS style.  

  1. Authors must add the conclusion section.

Response: It is really true as Reviewer suggested that we added the conclusion section in article. (5. Conclusions. Line 483-495)

Reviewer 2 Report

The manuscript by Xing et al. characterizes SiPHR1 gene as a key regulator in response to low phosphate in foxtail millet. To understand the molecular mechanisms under low phosphate conditions, the authors investigated chlorophyll and anthocyanin contents, enzymatic activity of purple acid phosphatase and phytase, antioxidant systems, and comprehensive gene expressions by RNA-Seq. Identification of significant DEGs and the constructed co-expression networks suggested SiPHR1 as a candidate gene for low phosphate response in this species. Further functional analysis of this gene demonstrated that SiPHR1 was involved in the response. This work was well performed. However, I have some comments for the current version of the manuscript as follows.

Abstract

Please explain all the abbreviations, for example, LP, CAT, POD, and SOD.

Figures

1)    Please explain the error bars in Figure 4. What kinds of statistical tests did you use? Please describe it.

2)    Figure 5C and 5D: The resolution of these figures is too low. I cannot see any node name.

Materials and Methods

1)    The authors should cite the latest papers about all the software tools and databases used in this study. For example, hisat2 and clusterprofiler.

2)    Please describe all the parameters used in the data analysis.

3)    To make the raw data publicly available is important for transparency, reproducibility, and reusability. Therefore, I would suggest that you should make all the raw data (short-read data) available in a public database, such as NCBI SRA.

See above.

Author Response

Dear Editors and Reviewers:

Thank you for your letter and for the reviewers’ comments concerning our manuscript entitled “ The functional characteristic of SiPHR1 as a hub gene responded for low phosphate via comparative transcriptomic and the co-expression network analysis in foxtail millet ”(ID: IJMS 2469279). Those comments are all valuable and very helpful for revising and improving our paper, as well as the important guiding significance to our researches. We have studied comments carefully and have made correction which we hope meet with approval. Revised portion are marked in color in the paper. The main corrections in the paper and the responds to the reviewer’s comments are as following:

Reviewer 2

Abstract:

Please explain all the abbreviations, for example, LP, CAT, POD, and SOD.

Response: As Reviewer suggested that all the abbreviations in the abstract has been explained. (Line 17-24)

Figures

1)    Please explain the error bars in Figure 4. What kinds of statistical tests did you use? Please describe it.

Response: Figure 4 displayed the expression of 13 candidate genes. The RNA-seq data and qRT-PCR data were used to verify mutually. The error bar in the figures means the Standard error which derived from the 3 replicates. The t-test analysis was used for statistical test.

2)    Figure 5C and 5D: The resolution of these figures is too low. I cannot see any node name.

Response: We have made correction according to the Reviewer’s comments, and the Figure 5 has been modified to make the node name in figure clear to see.

Materials and Methods

  • The authors should cite the latest papers about all the software tools and databases used in this study. For example, hisat2 and clusterprofiler.

Response: The software tools and databased used in this study have been cited or adressed in the manuscript and listed in reference. (Line 435-438 ). Such as reference 37, 39, 61.

  • Please describe all the parameters used in the data analysis.

Response: As Reviewer suggested that we describe the parameters used in the data analysis.

  • To make the raw data publicly available is important for transparency, reproducibility, and reusability. Therefore, I would suggest that you should make all the raw data (short-read data) available in a public database, such as NCBI SRA.

Response: Special thanks to you for your good comments. We have submitted all the raw data available in NCBI SRA (Our submission ID: SUB13720581). And all of the raw data will be released before 2024-07-31. 

Round 2

Reviewer 1 Report

Dear Chief Editor,

Thank you for providing the opportunity to review the revised manuscript. The authors have addressed all comments and incorporated changes suggested by reviewers during the first round of revisions. The revised version of the manuscript is improved as expected. Based on these revisions, now this study is a suitable contribution to the IJMS. I recommend the manuscript for publication.

Thank you

With best regards

Dear Chief Editor,

Thank you for providing the opportunity to review the revised manuscript. The authors have addressed all comments and incorporated changes suggested by reviewers during the first round of revisions. The revised version of the manuscript is improved as expected. Based on these revisions, now this study is a suitable contribution to the IJMS. I recommend the manuscript for publication.

Thank you

With best regards